# Characterization and On-Field Performance of the MuTe Silicon Photomultipliers

**Jesús Peña-Rodríguez [1,*] , Juan Sánchez-Villafrades [2] , Hernán Asorey [3,4] and Luis A. Núñez [1,5]**

1   Escuela de Física, Universidad Industrial de Santander, Bucaramanga 680002, Colombia
2   Escuela de Ingeniería Eléctrica, Electrónica y de Telecomunicaciones, Universidad Industrial de Santander, Bucaramanga 680002, Colombia
3   Departamento Física Médica, Centro Atómico Bariloche, Comisión Nacional de Energía Atómica, Bariloche R8402AGP, Argentina
4   Instituto de Tecnologías en Detección y Astropartículas, Buenos Aires CP B1650KNA, Argentina
5   Departamento de Física, Universidad de Los Andes, Mérida 5101, Venezuela
*   Correspondence: jesus.pena@correo.uis.edu.co

**Abstract:** The Muon Telescope, MuTe, is an instrument for imaging volcanoes in Colombia. It consists of a scintillator tracking system and a water Cherenkov detector for particle energy measurement. The Muon Telescope operates autonomously in high-altitude environments where the temperature gradient reaches up to 10 °C. In this work, we characterize the telescope silicon photomultipliers' breakdown voltage, gain, and noise for temperature variations spanning 0 to 40 °C. We demonstrate that the discrimination threshold for the Muon Telescope hodoscope must be above 5 photo-electrons to avoid contamination due to dark count, crosstalk, and afterpulsing. We also assess the detector counting rate depending on day-night temperature variations.

**Keywords:** Muography; Silicon Photomultipliers; Dark Count; Crosstalk; Afterpulsing

## 1. Introduction

Muography is a non-invasive technique for imaging anthropic and geologic structures [1–12] by measuring the crossing muon flux using sensitive hodoscopes made of nuclear emulsions [2,13], gaseous chambers [14–17] and scintillators [4,8,11,18–20]. Scintillation hodoscopes provide flexibility in the implementation, low cost, and robustness against environmental variables such as humidity, temperature, and atmospheric pressure [21]. When an ionizing particle interacts with the scintillator crystal lattice, it knocks electrons out from the valence band to bound states called excitons. These excitons emit photons in the near-ultraviolet spectrum due to recombination by de-excitation. Some dopants are added to the primary scintillation material so as to obtain a light in a longer wavelength, with the view that the absorption length of the ultraviolet light is relatively short. The resultant emission range of the scintillator mismatches the sensitivity of the majority of the photosensors necessary for adding a wavelength-shifting fibre [22].

Silicon photomultipliers (SiPMs) offer a solution for high granularity hodoscopes to be deployed in volcanic areas because of their small dimensions, robustness, and low power consumption [23]. SiPMs contain a dense array of small photon avalanche diodes operating in Geiger mode. When a photon interacts with a SiPM microcell, an avalanche process starts generating a photocurrent flowing through a quenching resistor, which causes a dropoff of the diode bias below the breakdown value preventing further Geiger-mode avalanches. The electrical pulses generated by the SiPM are directly related to the number of incident photons.

SiPMs are used in medical imaging, particle physics, and high-energy astrophysics. SiPM characterization includes gain, overvoltage, and noise parameterization in darkness, using LED or radioactive sources [24,25]. In muography, SiPMs have replaced multianode

photomultipliers due to their low noise and electromagnetic field robustness. The main drawback of SiPMs is that performance parameters such as gain, photodetection efficiency, and breakdown voltage are susceptible to temperature variations. Muon telescopes operate outdoors under uncontrolled temperature conditions, unlike applications where detectors work under well established temperatures. Thermo-electric cells can control temperature of muon telescope SiPMs, but this methodology carries an increase in power consumption, which reduces the powering efficiency of autonomous muon telescopes [23].

This paper analyses the characterization of the Muon Telescope (MuTe) SiPMs breakdown voltage, gain, dark count, crosstalk, and afterpulsing depending on temperature and over-voltage. Section 2 shows a technical description of the MuTe. Section 3 describes the experimental setup and the data acquisition system for the SiPM characterization measurements. Section 4 shows the breakdown voltage, gain, and noise parameterization results. Section 5 presents the temperature conditions at the Cerro Machín volcano and their effect on the MuTe mechanical structure, and Section 6 exhibits the dependence between the flux and the temperature of the MuTe tracking system operating outdoors. Section 7 summarizes conclusions and remarks.

## 2. The Muon Telescope

The MuTe is a hybrid detector composed of a hodoscope and a Water Cherenkov Detector (WCD), which will be installed in one of the most dangerous volcanoes in Colombia, the Cerro Machín, located at 2750 m.a.s.l. on the Cordillera Central near to the municipality of Cajamarca [26,27]. The MuTe hodoscope consists of two scintillation panels, each of $30 \times 30$ strips 120 cm length, and 4 cm wide. Each strip has a 1.8 mm hole for a multi-cladding wavelength shifting (WLS) fibre (Saint-Gobain BCF-92) with 1.2 mm diameter, an absorption peak at 410 nm and an emission peak of 492 nm [28]. The WLS fibre is coupled to a silicon photomultiplier (SiPM, Hamamatsu S13360-1350CS) [29]. The SiPM has a photosensitive area of $1.3 \times 1.3$ mm$^2$, 667 pixels, a fill factor of 74%, a gain from $10^5$ to $10^6$ and a photon-detection efficiency of 40% at 450 nm.

## 3. Experimental Setup

### 3.1. Dark Current

The experimental setup to measure the SiPM dark current as function of temperature is indicated in Figure 1. The SiPM is placed on an isolated aluminium holder whose temperature is controlled by two Peltier cells (TEC1-12706 from Hebei I.T.) and measured using an LM35 sensor (linear 10 mV/$^\circ$C scale factor and $-55\,^\circ$C to 150 $^\circ$C range). A proportional–integral–derivative (PID) control (implemented in a microcontroller Atmega328) generates two pulse-width-modulated signals whose duty cycle depends on the control error. The error is the difference between the measured temperature and the pre-established set-point. The control signals drive the direction (cooling or heating) and amplitude (fast or slow) of the current flowing through the Peltier cells using an H-bridge with an optically coupled isolator circuit.

A C11204 power module biases the SiPM ranging in voltage from 40 V to 60 V. The dark current is measured by a 2 nA accuracy picoammeter. The SiPM bias voltage and temperature are recorded individually by a 10-bit analogue-to-digital converter (ADC) with a sampling rate of 1 Hz. All the setup components are placed inside a grounded dark box to avoid external light contamination and electromagnetic interference.

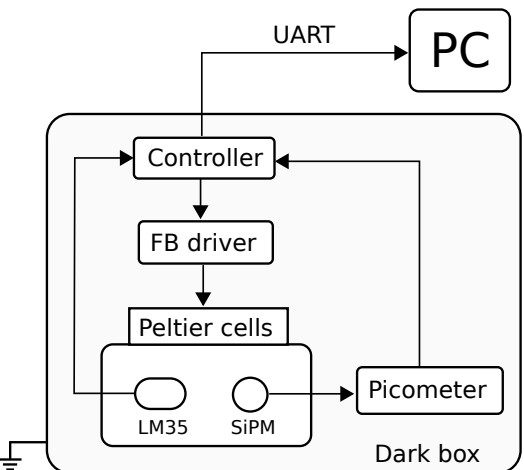

**Figure 1.** Experimental setup for measuring the SiPM dark current in darkness conditions. The SiPM is positioned in the aluminium holder inside the dark box. A PID controller sets the holder temperature via two Peltier cells with a full bridge (FB) driver. The controller sends the data to a computer via a UART (universal asynchronous receiver–transmitter) port.

### 3.2. Gain and Noise

A second experimental setup is used to estimate the SiPM gain and noise at several temperatures and voltages with pulse light stimulating. The light source must fulfil two features: a wavelength matching the SiPM spectral sensitivity and a pulse width of the order of a few ns [30,31].

The light pulser generates an ultra-short ($<$10 ns) 480 nm pulse with a frequency of 500 Hz. A 50 cm WLS fibre (Saint-Gobain BCF-92) transports the light towards the SiPM; simultaneously, a square signal triggers the DAQ system. The signals generated by the SiPM are amplified 94 times using a low noise current feedback operational amplifier (OPA691 from Texas Instruments) and digitized by a Red Pitaya ADC channel with a sampling frequency of 125 MHz and 14-bit resolution. A diagram of the setup is shown in Figure 2.

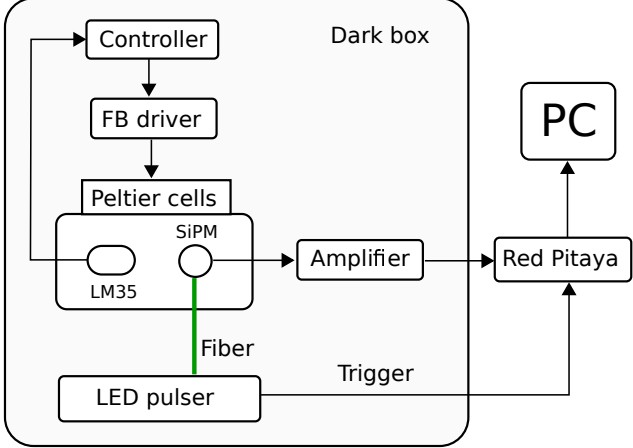

**Figure 2.** Experimental setup for measuring the gain and noise of the SiPM under stimulated conditions. The SiPM is stimulated by a 480 nm pulsed light of $\sim$10 ns width at 500 Hz. The SiPM signal is digitized by the Red Pitaya at 14-bit/125 MHz.

## 4. SiPM Calibration

### 4.1. Breakdown Voltage

The breakdown voltage ($V_{br}$) is the point where the SiPM enters Geiger mode. Such a point can be established using several methods [32]. In this case, we use the tangent method, which consists of finding the interception between a tangent line fit to the IV (dark

current vs bias voltage) curve and the baseline. Figure 3 shows the SiPM IV curve at 25 °C where the $V_{br}$ was at ~52.3 V.

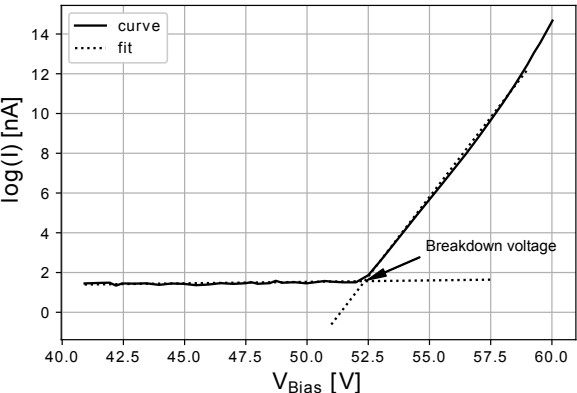

**Figure 3.** Breakdown voltage value found using the tangent method for the IV curve of SiPM operating at 25 °C. The $V_{br}$ (52.3 V) is located at the intersection between the fit and the baseline.

We measured the IV curves from 40 V to 60 V for temperatures between 0 °C and 40 °C with 5 °C step as shown in Figure 4 (left panel) using the breakdown voltage experimental setup. In the Geiger region, the IV slope increases with the temperature reaching a dark current above 400 nA at 40 °C. The breakdown voltage has a linear relation with temperature decreasing with a ratio of 41.7 mV/°C as is shown in Figure 4 (right panel). In on-field applications, an adaptive bias voltage to compensate for temperature changes in the SiPM could be taken into account to guarantee a stable gain and low noise levels [31].

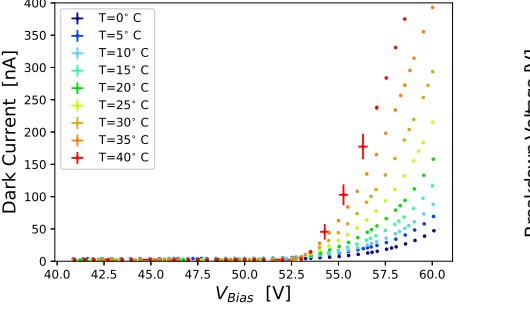
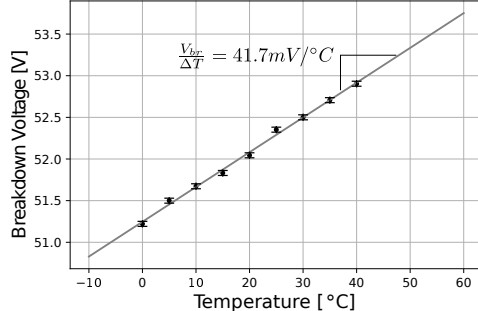

**Figure 4.** Temperature dependence of the SiPM breakdown voltage. (**Left**): IV curves ranging from 0 °C to 40 °C. (**Right**): Variation of the breakdown voltage $V_{br}$ with temperature.

### 4.2. Charge Spectrum and Gain

The gain of a SiPM microcell is defined as the ratio of the output charge to the charge of an electron *e* [33]. The output charge can be calculated as,

$$Q = \frac{Q_{ADC} V_{ADC} \Delta_t}{R G_a}, \tag{1}$$

where $Q_{ADC}$ is the digitized area under the pulse, $V_{ADC}$ is the equivalent voltage for one ADC unit, $\Delta_t$ is the digitization time step, $R$ is the input resistor and $G_a$ the gain of the electronics front-end. In Figure 5, the charge spectrum of the SiPM operating at 56 V and 25 °C is shown.

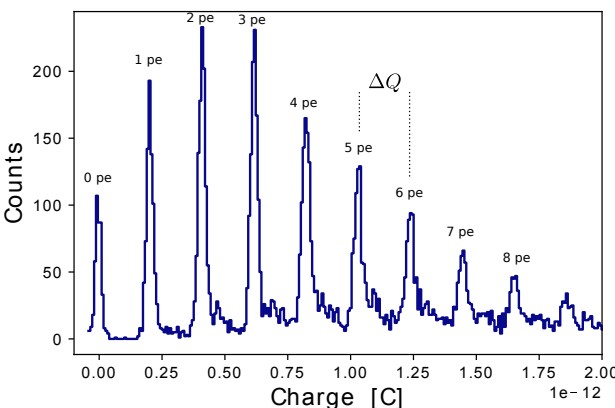

**Figure 5.** Charge spectrum of the SiPM operating at 56 V/25 °C. The first peak is the pedestal and the following represent the photoelectron equivalents. The inter-peak charge $\Delta Q$ determines the SiPM gain.

The average separation between two adjacent peaks, $\Delta Q$, in the charge histogram corresponds to the charge from a single Geiger discharge. This can be used to accurately calculate the gain $G$ as follows:

$$G = \frac{\Delta Q}{e}. \tag{2}$$

The SiPM gain depends on the bias voltage ($V_{\text{bias}} = V_{\text{br}} + \Delta V$), the higher the bias voltage, the higher the gain. To estimate the gain dependence on the over-voltage ($\Delta V$) in the SiPM S13360-1350CS, we measured three charge spectra for $\Delta V$ = 1.7 V, 2.7 V, and 3.7 V at 25 °C. Figure 6 shows the charge spectra (left panel) and the estimated gain (right panel) for these cases.

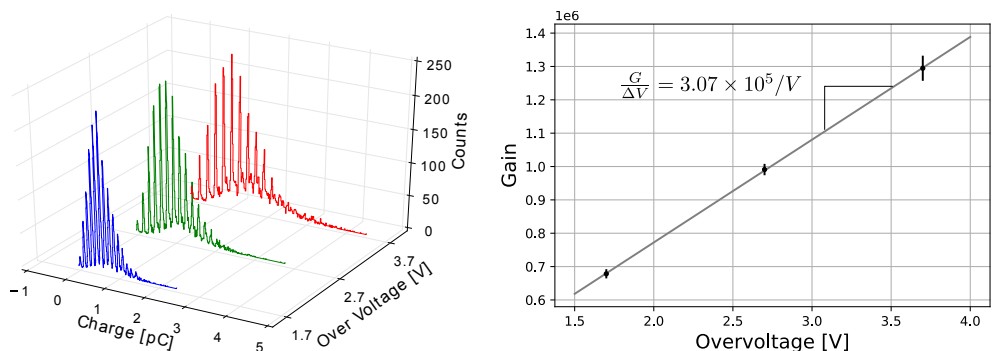

**Figure 6.** (**Left**): Charge spectrum for $\Delta V$ = 1.7 V (blue), 2.7 V (green), and 3.7 V (red). (**Right**): Measured gain variation ratio depending on the over-voltage.

The separation between charge peaks grows as the over-voltage increases—indicating an increment in gain. The gain change ratio was estimated $\sim 3.07 \times 10^5$/V, i.e., for $\Delta V$ = 1.7 V ($V_{bias}$ = 53 V) the gain is roughly $0.7 \times 10^6$ and for $\Delta V$ = 3.7 V ($V_{bias}$ = 56 V) the gain is $1.3 \times 10^6$.

### 4.3. Photo-Electron Spectrum

The output pulse amplitude from SiPMs is proportional to the number of incident photons based on the fact they are made of an array of APDs connected in parallel. The photo-electron (pe) spectrum determines the equivalent value (voltage or current) of a photon interacting with the active area of the SiPM. This value establishes the threshold for measuring dark count rate (DCR), crosstalk and afterpulsing noise.

In Figure 7, the persistence histogram (left-panel) of the pulse shape and the peak histogram (right-panel) for $10^4$ pulses at 56 V/25 °C are shown.

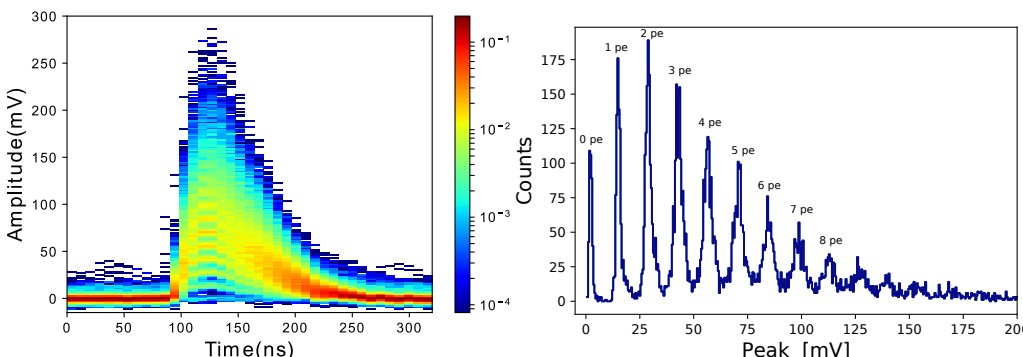

**Figure 7.** (**Left**): Waveform of the Hamamatsu S13360-1350CS under stimulation. (**Right**) Photo-electron spectrum resulting from integrating the area under pulse over a time window of 300 ns.

The histograms reveal that pulses of 1 pe and 2 pe have more probability of occurrence than others. These pulses are mainly generated by the SiPM noise [33]. The resulting equivalent voltage for 1 pe is ∼13.5 mV, therefore the threshold for measuring the SiPM DCR must be set below 13.5 mV and for the cross-talk below 27 mV.

### 4.4. Noise

SiPMs are affected by correlated noise (crosstalk and afterpulsing) and non-correlated noise (DCR) [34]. These noise sources impose a lower measurement limit in SiPM-based experiments. We performed a noise analysis of the MuTe SiPMs taking into account their temperature and over-voltage dependency.

### 4.4.1. Dark Count Rate

The main source of noise in SiPMs is the DCR. It appears as a consequence of avalanches processes fired by electrons thermally generated in the silicon crystal. Signals generated by thermal electrons and single photons are identical. The DCR is measured under dark conditions by counting events above a 0.5 pe threshold.

The DCR is calculated as follows,

$$\text{DCR} = \frac{N_{1\text{pe}}^{\text{B}}}{T^{\text{B}} N_{\text{p}}}, \tag{3}$$

where $N_{1\text{pe}}^{\text{B}}$ is the number of events above 0.5 pe before stimulation and $N_{\text{p}}$ is the total number of recorded events.

We measured the DCR for different thresholds spanning from 0.1 pe to 3.1 pe at 56 V/25 °C as shown in Figure 8. The resulting curve has a stepped shape because of the amplitude discretization of the SiPM pulses. At 0.5 pe, the DCR is ∼$2 \times 10^5$ Hz in agreement to the expected value provided by the SiPM S13360-1350CS datasheet which range between $0.9 \times 10^5$ Hz and $2.7 \times 10^5$ Hz.

The DCR drastically decreases while the measurement threshold increases. We found a DCR of $9 \times 10^3$ Hz at 1.5 pe, $6 \times 10^2$ Hz at 2.5 pe, and <10 Hz at 3.5 pe.

To characterize the DCR as a function of the over-voltage, we carried out DCR measurements for three cases (1.7 V, 2.7 V and 3.7 V) at 25 °C. Figure 9 shows that the DCR increases with a slope ∼11.16 kHz/V.

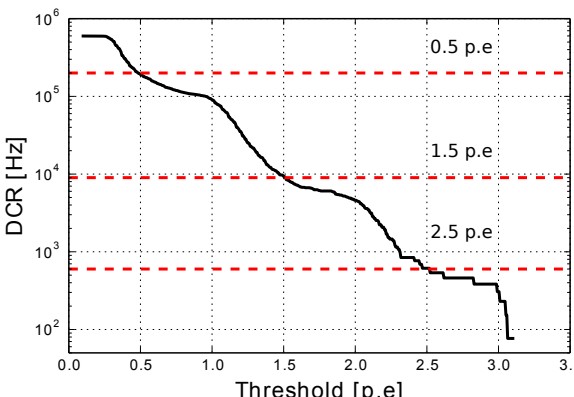

**Figure 8.** Dark count rate as a function of the detection threshold. The curve shape presents three breaks at 1 pe, 2 pe and 3 pe because of the discretization effect on the pulse amplitude.

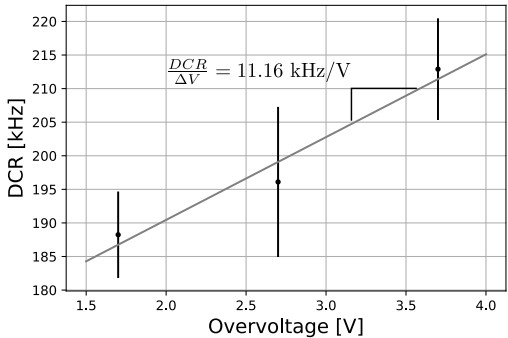

**Figure 9.** Dark count rate as a function of the over-voltage ranging from (1.7 V to 3.7 V) at 25 °C.

The DCR correlation with the SiPM temperature was also evaluated. We estimated a ratio 0.85 kHz/°C after analyzing DRC measurements from 0 °C to 40 °C at 56 V.

### 4.4.2. Afterpulsing and Crosstalk

Afterpulsing is generated by trapped electrons in silicon impurities during an avalanche process. These electrons are released a few nanoseconds later, creating new avalanches—i.e., consecutive pulses [35]. The amplitude of afterpulses increases with the retention time of the trapped electron.

The afterpulsing probability $P_{AP}$ is calculated as follows

$$P_{AP} = \frac{N_{1pe}^{A} - N_{1pe}^{B}}{N_{P}} \times 100, \tag{4}$$

where $N_{1pe}^{A}$ and $N_{1pe}^{B}$ are the number of events with amplitude $> 0.5$ pe after and before stimulation with the light pulser.

Crosstalk occurs when charge carriers (inside the avalanche) emit photons that interact with neighboring cells triggering secondary avalanches.

The crosstalk probability [36,37] is defined as

$$P_{CT} = \frac{N_{2pe}^{B}}{N_{1pe}^{B}} \times 100, \tag{5}$$

where $N_{2pe}^{B}$ is the number of events with amplitude $> 1.5$ pe before stimulation.

In Figure 10, the afterpulsing and crosstalk versus the SiPM over-voltage are shown. Both increase exponentially with the over-voltage, with the crosstalk greater than the afterpulsing, which reaches 3% at 56 V/25 °C with the crosstalk reaching 5%.

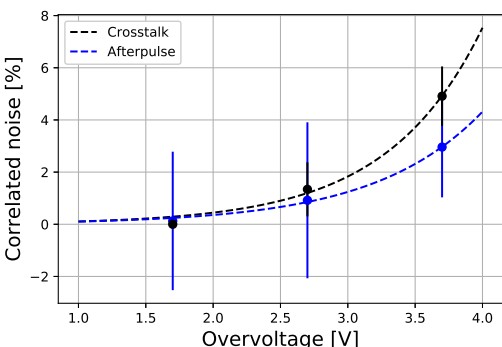

**Figure 10.** SiPM crosstalk (black line) and afterpulsing (blue line) depending on the over-voltage.

The correlated noise dependency on temperature was analyzed by performing afterpulsing and crosstalk measurements from 0 °C to 40 °C at 56 V. At 0 °C, the probability of afterpulsing is below 2% and of crosstalk is below 4%. The afterpulsing increases faster than crosstalk with temperature, rising up almost 5% at 40 °C while crosstalk reaches 6%.

To reduce the noise caused by dark count, crosstalk, and afterpulsing, we concluded that the minimum discrimination threshold for the MuTe scintillator hodoscope must be above 5 pe. The breakdown voltage shifting due to temperature variations will cause a modulation of the detection rate. This can be solved using closed-loop control of the SiPMs bias voltage or corrected latter in the offline data analysis.

## 5. Operating Temperature Conditions of MuTe

### 5.1. Weather at the Cerro Machín Volcano

The Cerro Machín volcano has the typical weather conditions of the Andean mountains in Colombia. According to the Colombian Hydrology, Meteorology and Environmental Studies Institute (IDEAM), the average temperature at Cerro Machín is 16 °C, the relative humidity 85%, and the maximum wind speed 30 m/s. During the rainy season, the temperature drops to 0 °C and during the dry season it rises to 25 °C. The rainy season occurs between April to May and October to November, and the dry season is usually from December to January and July to August. The day–night temperature gradient at the Cerro Machín volcano is around 10 °C along the dry and rainy seasons.

### 5.2. Heat Transfer in the MuTe Structure

We computed a thermal analysis of the MuTe mechanical structure using the SOLID-WORKS CAD SOFTWARE. The heat sources were: the environmental temperature (16 °C), solar radiation (4.5 kWh m$^{-1}$day$^{-1}$), cooling by wind (30 m/s), and heating by the electronics power consumption (12.5 W). We also inputted the thermal features of the metallic chassis supporting the WCD and the hodoscope [27].

In Figure 11, the temperature distribution on the MuTe structure resulting from the thermal simulation is shown. The direct incidence of the solar radiation (solid arrow) causes a maximum temperature of 60 °C in the middle of the scintillation panels, but this drops to 26 °C due to the convection created by the frontal wind (dashed arrow). The water volume inside the WCD dissipates the heat of the stainless steel cube. The maximum temperature on the WCD is ∼40 °C.

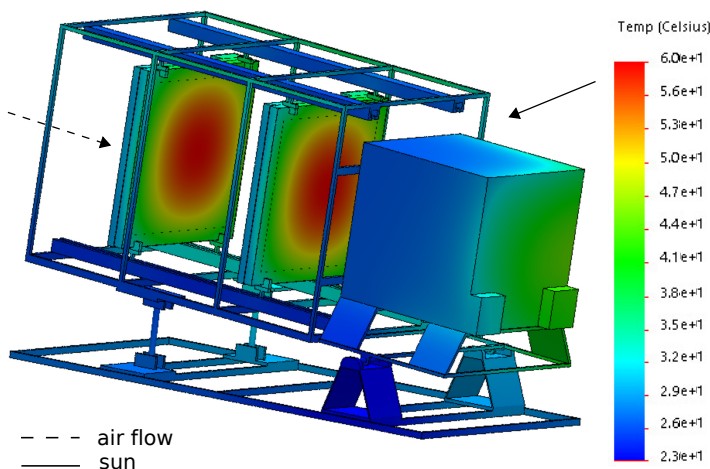

**Figure 11.** Heat distribution of the MuTe structure under the environmental conditions at the Cerro Machín volcano. The solid arrow represents the incident solar radiation while the dashed arrow indicates the wind direction. The maximum temperature at the centre of the scintillation panels reaches 60 °C.

## 6. Temperature Influence on the MuTe-SiPMs

In this section, we analyze how temperature affects the SiPM parameters under real observational conditions. This procedure uses the characterization ratios presented in Section 4 and temperature measurements.

We use temperature data recorded at the Cerro Machín volcano during the 2017 rainy season between 22–23 November. The day–night temperature cycle starts/ends at 00:00 h with ∼10 °C. The temperature drops to a minimum value of ∼8.5 °C at 06:30 h and rises to a maximum of ∼14.5 °C at 13:00 h, as shown Figure 12.

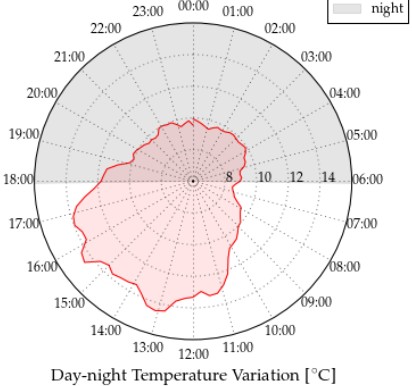

**Figure 12.** Day–night temperature cycle at the Cerro-Machín volcano during the rainy season (22–23 November). The gray shadow indicates the night period starting at 18:00 and ending at 06:00. The minimum temperature (∼8.5 °C) is recorded at 06:30 and the maximum (∼14.5 °C) at 13:00.

The estimated SiPM breakdown voltage and DCR along the day–night cycle are presented in Figure 13. The maximum temperature gradient is ∼6.1 °C which represents a breakdown voltage (41.7 mV/°C) deviation of ±126 mV from the nominal value (53.2 V). This breakdown shift affects the SiPM gain ($3.07 \times 10^5$/V) causing a deviation ∼$0.8 \times 10^5$.

As the temperature of the SiPM increases, the number of thermally generated electrons on the silicon material also increases. The DCR absolute variation is ∼5.2 kHz. We can expect the DCR to vary between 207 kHz and 212.2 kHz, assuming the SiPM operates at 56 V where the nominal DCR is approximately 210 kHz.

The maximum variance of the pulse amplitude is about 0.8 mV for a $\Delta T \sim 6.1$ °C which represents roughly 6% of the voltage separation between two consecutive photoelectrons

(∼13.5 mV at 56 V). Figure 14 shows the resulting variance around the threshold voltage (right) and its respective photoelectron equivalent (left).

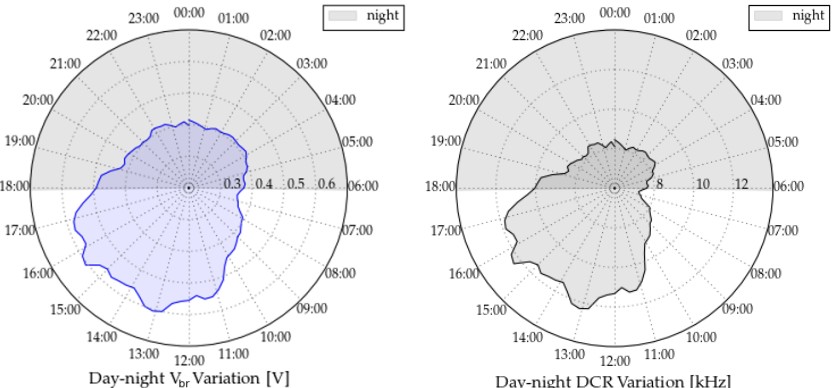

**Figure 13.** MuTe SiPM breakdown voltage (**left**) and DCR (**right**) variation as a function of the temperature values at the Cerro Machín volcano.

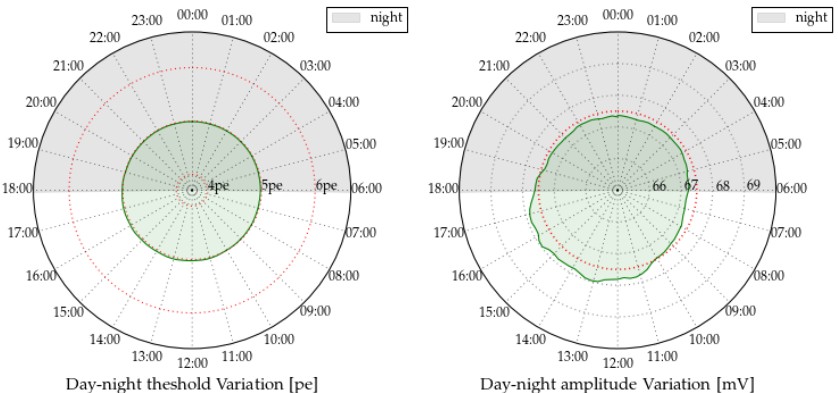

**Figure 14.** Photo-electron (**left**) and pulse amplitude (**right**) variation of the MuTe-SiPM with temperature values at the Cerro Machín volcano.

We analyzed four days of data from 20 December 2019 to 24 December 2019 in order to evaluate the MuTe-SiPMs behaviour in field conditions. We set a hodoscope discrimination threshold of 8 pe (108.5 mV) so as to suppress the noise contributions due to DCR, after-pulsing, and crosstalk ($<10^{-4}$ Hz) and allowing the signals from minimum ionizing muons (∼12 pe for a distance SiPM-WLS fibre of 0.5 mm) [38]. Figure 15 shows the in-coincidence detection rate, and temperatures of the rear ($T_R$) and frontal ($T_F$) panels. For this measurement, MuTe was set pointing towards the horizon, with an angular aperture of 52°, and an inter-panel separation of 2.5 m.

The panel temperature oscillates from 20 °C to 30 °C, representing a gradient of 10 °C. A 10 °C gradient represents a variation in the pulse amplitude around 14.8%. However, this variation increases the breakdown voltage ∼417 mV, reducing the overvoltage and the SiPM gain causing a reduction of the detected rate. The measured average flux is ∼3.1 events/s varying roughly 11.2% between the maximum and minimum temperatures. We estimated the temperature dependency (−0.057 Hz/°C) of the muon flux recorded by the telescope. The muon flux correction is carried out by subtracting the product between the instantaneous temperature and the dependency coefficient from the instantaneous muon flux [39].

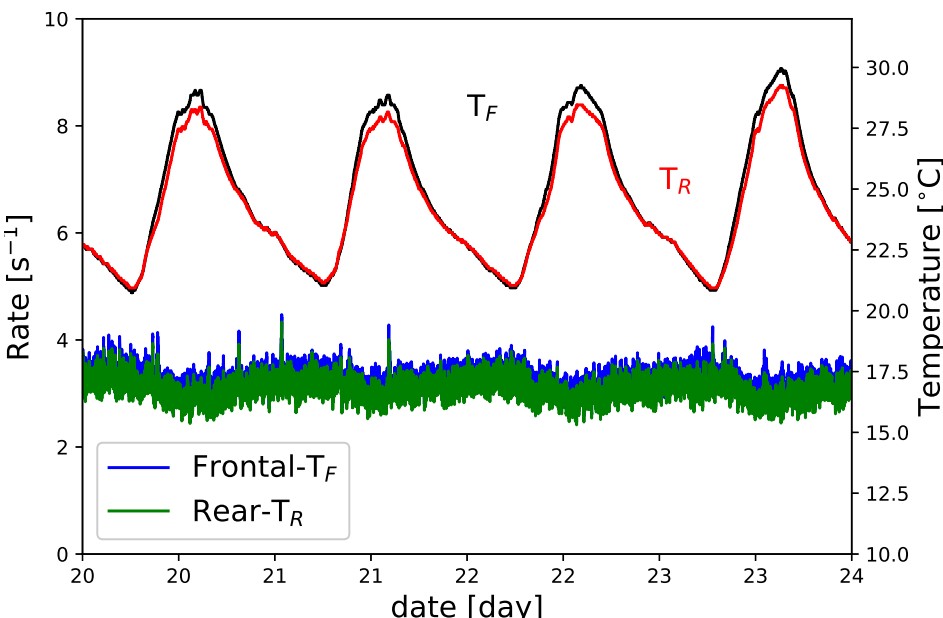

**Figure 15.** In-coincidence hodoscope rate modulation depending on the environmental temperature of the MuTe recorded from 20 December 2019 to 25 December 2019. The green line displays the rear panel rate under temperature $T_R$ and the blue line the frontal panel rate under temperature $T_F$.

## 7. Discussion and Conclusions

We evaluated the SiPM S13360-1350CS breakdown voltage, gain, and noise depending on the over-voltage and temperature. Temperature tests ranged from 0 °C to 40 °C covering the temperature spectrum of the observation site at Cerro Machín Volcano, Colombia. The SiPM breakdown voltage variation ratio was about 41.7 mV/°C, indicating a pulse amplitude shift of 14.8%, which is not indicative of jumping between photoelectron levels. We also estimated a gain increase ratio of about $3.07 \times 10^5$/V for over-voltage changes on the SiPM.

With the noise characterization, we found that dark count rate decreases by several orders of magnitude (<100 Hz) at a threshold above 3 pe and DCR increases with a ratio of 11.16 kHz/V as a function of the SiPM over-voltage. The afterpulsing and crosstalk probabilities show a non-linear growth with the temperature reaching up to 3% and 5%, respectively, for an over-voltage of 3.7 V. We recommend a minimum discrimination threshold of 5 pe so as to reduce correlated and non-correlated noises in the SiPMs of MuTe.

Environmental temperatures modulate the hodoscope rate in the field test, reaching a maximum deviation of 11.2%. The modulation shows an inverse correlation with SiPM's temperature (−0.057 Hz/°C) due to the increase in the breakdown voltage and the SiPM gain reduction.

**Author Contributions:** Conceptualization, J.P.-R.; methodology, J.S.-V. and J.P.-R.; investigation, J.S.-V. and J.P.-R.; supervision, H.A. and L.A.N.; writing—review and editing, J.S.-V., J.P.-R., H.A. and L.A.N. All authors have read and agreed to the published version of the manuscript.

**Funding:** This work was supported by the Departamento Administrativo de Ciencia, Tecnología e Innovación of Colombia (ColCiencias) under contract FP44842-082-2015 and to the Programa de Cooperación Nivel II (PCB-II) MINCYT-CONICET-COLCIENCIAS 2015, under project CO/15/02.

**Data Availability Statement:** Not applicable.

**Acknowledgments:** The authors acknowledge the Grupo Halley Laboratory and Universidad Industrial de Santander where SiPM measurements and the Muon Telescope installation were Tperformed.

**Conflicts of Interest:** The authors declare no conflict of interest.

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
