# Peer review of "Characterization and On-Field Performance of the MuTe Silicon Photomultipliers"

_instruments, doi:10.3390/instruments7010007_

Round 1
Reviewer 1 Report
The authors show a characterization study of the SiPM used in the MUTe hodoscope and describe different measurements performed for temperature dependence of breakdown voltage, gain and noise. The results are of interest for researches using scintillation detectors readout by SiPM´s and therefore are worth for publication, however some corrections are needed as described below.
General comments:
1. I think it is important to cite other authors who have performed studies on SiPM characterization and discuss differences with their results, I just give some examples but authors can do a more thorough search and provide a more detailed description of the work performed in the similar line of studies as theirs:
https://www.mdpi.com/1424-8220/21/17/5947
https://link.springer.com/chapter/10.1007/978-981-19-2354-8_146
https://journals.jps.jp/doi/10.7566/JPSCP.27.011003
2. Some measurements are indicated without including any information about their uncertainties. Fits are performed on some of these measurements but the quality and conclusions from the fits are arguable without a robust statistical analysis. For example, Figure 4 left panel, Figure 6 right panel, figures 9 and 10 have error bars missing. Also, the error bars displayed on Figure 4 right panel seem to be equal across all data points plotted.
3. The authors conclude that a threshold of 5 p.e. is needed to avoid contamination from dark counts, noise and cross talk, they even performed in situ measurements in Cerro Machin but they fail to provide measurements of number of p.e. from minimum ionizing muons going through their hodoscope and indicate how the measurements of the low atmospheric muon rates could be affected by any leftover noise above the 5 p.e. threshold chosen.
4. A thorough review of English grammar is needed through the whole article. There aresentences in English that need to be reviewed, also some typographic erros and few words such as “reunion” that shouldn’t belong to the text. I suggest to the authors to use any editing software to improve the English grammar of the document.
5. It is advised to the authors to check the number of auto-citations and to reduce them to the minimum, which should be one or two publications where a more detailed descripticon is performed about MUTE.
Specific comments:
Line 34: cite a reference instead of a footnote with a webpage link. You could cite references 36 and 37 instead.
Line 43: Please explain the reason for the choice of the SipM, and how it couples to the emission peak of the WLS fiber. Why not to select a SiPM with a better match to the spectral emission of the WLS fiber?
Line 57: What is the temperature range for the LM35 sensor to have a linear behavior? What is the accuracy of the temperature measurement?
Line 77: What is the motivation behind the specific gain of 94?
Line 85: Please explain how the IV curve was obtained. If the curve is obtained from your own measurements in the lab, please indicate the measurements and their uncertainties. The curve in Figure 3 is a continuous curve.
Line 88: There is a conflict the caption of Figure 3 and the text, in the document it is stated that they are measurements performed by the authors, while in the caption it says that the plots are from Hamamatsu.
Line 88: How is the breakdown voltage measured for high temperature cases were the IV curve is non-fully exponential. Which uncertainty is added to the non-exponential behavior of the IV curve when using the tangent method?
Line 103: Please specify in more detail how the gain is obtained? How do you measure the difference in charge peaks of the spectrum? Do you use the difference among the many peaks or one specific, as indicated in figure 5? Wouldn’t it be more accurate to perform a multi-gaussian fit to obtain Delta_Q?.
Line 107: Is it estimated gain or measured gain. Error bars are missing in Fig. 6 right panel.
Title of Figure 7, left panel: “Is this title really needed?, what is its meaning?
Lines 121-122: The statement that 1 pe and 2 pe are mainly generated by SiPM noise needs some physics justification.
Line 123: I think an important measurement of a MIP from cosmic muons is missing, this measurement compared the noise spectrum would be the one to allow to determine the proper threshold. Please provide arguments in this direction.
Line 144: Please provide an explanation why a DCR rate of 10 HZ in a single SiPM is not a problem, when compared to the expected rate from quasihorizontal atmospheric muons producing signal in one specific channel of MUTE.
Figure 9: Please provide error bars on the measured points. With no error bars it is difficult to understand the quality of the linear fits indicated.
Line 156: Please explain equation (4) in more detail, because it means meaurements of number of events with 1 pe in different time windows.
Line 159: Please provide the reason why cross talk only triggers secondary pulses with only 2 o 3 pe.
Figure 10: Please provide error bars for the indicated measurements.
Line 170: You need to provide how many pe you expect from a minimum ionizing muon incident to the detector. In this way, the reader can compare this threshold with a possible reduction in efficiency of the system.
Line 181: Please provide a reference where readers can find more information about weather conditions of Cerro Machin.
Line 210: The DCR reported is too high compared to the expected muon rate, which is not mentioned in the text. Have you performed muon rate measurements in situ?
Line 215: Here you indicate that the threshold is 8 pe, in other places of the document ypu indicate that this number is 5 pe.
Figure 15: Is muon rate, form coincidence of the two hodoscope panels what is plotted in this figure?
Line 229: Could you please explain the value reported in this line and a few extra words about the correction is performed offline?
Line 232: Please provide an explanation why you only tested one specific siPM and did not consider other candidates.
English comments/suggestions:
Line 1: “The Muon Telescope is a muography experiment” Muon telescope here souns as too generic, I would use the name of the project: MUTE (Muon Thelescope) is a muography experiment…
Line 6: “We demonstrated that the discrimination threshold for the MuTe hodoscope”-> “we demonstrate..”
Line 21: “the sensitivity of the most photosensors” -> “the sensitivity of most of the photosensors”
Line 27: “which causes that the diode bias drops below “ ->” which causes a dropoff of the diode bias below”
Line 30: “performance parameters like gain..” -> “performance parameters such as gain..”
Line 40: “The WLS fiber is coupled, with a silicon” -> “ The WLS fiber is coupled to a silicon”
Line 49: “volcano and their affectation on the MuTe” -> “volcano and their effect on the MuTe”
Line 54: No mention about different experimental setups have been made so far up to this point of the document, so please rephrase properly: “The experimental setup use to measure the SiPM dark current as function of temperature is indicated in Fig. 1.”
Line 59: “pulse-width-modulate signals” -> “pulse-width-modulated signals”
Caption Figure 1: “by means a PID” -> “by means of a PID”. What is the meaning of UART and FB in the schematics of Figure 1?
Line 70: “In the second experimental setup, we estimate the SiPM gain” -> “A second experimental setup is used to estimate the SiPM gain”
Line 74: “The light pulser generates an ultra-short (< 10 ns) 480nm light pulse”: the word “light” appears twice in the same sentence.
Line 96: “to the charge 96 on an electron” -> “to the charge fn an electron”
Line 140: “corresponding with the expected value” -> “in agreement to the expected..”
Line 156: “the trapped electron. Reunion” -> please correct this sentence.
Line 216: “to take reuniónout the noise contributions” -> please correct this sentence
Line 234 : “Temperature testes ranged from” something is wrong with this sentence, please correct.
Line 239: “several magnitude orders” -> “several orders of magnitude”
Author Response
We thank the detailed suggestion from the referees, which improves the readability of our manuscript. Attached you will find the revised version of the preprint instruments-2070854 (Characterization and on-field performance of the MuTe Silicon Photomultipliers).

Reviewer 2 Report
GENERAL OBSERVATIONS
________________________________________
* Avoid acronyms in abstract
* Mix when using of passive and first person
* Mix of labeling left and right panels in figure captions.
(Left): -> Left:
(Right): -> Right:
* Subscript must be upright:
V_{br} -> V_\mathrm{br}
V_{ADC} -> V_\mathrm{ADC}
G_a -> G_\mathrm{a}, etc
IN-LINE CORRECTIONS
________________________________________
ABSTRACT
3. The Muon Telescope operates...
5. SiPM -> silicon photomultipliers.
6. of the Muon Telescope silicon photomultipliers... 5 photo-electrons.
1. INTRODUCTION
21. the most -> the majority
22. (Photomultiplier or SiPMs): unnecessary. Need to define SiPM acronym.
22. to add -> the usage of a / using a / adding a.
23. SiPMs -> Silicon photomultipliers (SiPMs).
32. remove coma.
34. Reference of MuTe instead of footnote?
46-48. Mix using passive and first person. keep one (preferably passive).
2. EXPERIMENTAL SETUP
57. measured -> connected / coupled
64. remove SiPM model (S13360-1350CS)
Figure 2. remove SiPM model
3. SiPM CALIBRATION
85. fitted -> fit
86. we show -> is shown (first person -> passive)
87. was found "at"...
89. left-panel -> left panel
92. right-panel -> right panel
Figure 3. remove SiPM model from caption
Figure 4. remove SiPM model from caption
Figure 4. (Left): / (Right): -> Left: / Right:
Figure 4. V_{br} variance ratio depending on the temperature -> Variation of the breakdown voltage with temperature.
Equation 1. add coma at the end.
99. Using T for time can be misleading (in particular when T is also referred to temperature). suggestion of using "t", i.e., \Delta \mathrm{t} instead.
Figure 5. remove SiPM model from caption
103. remove coma
Equation 2. add dot at the end
107. left-panel / right-panel -> left panel / right panel
Figure 6. \Delta \mathrm{V}
116. photoelectron -> photo-electron (pe)
Figure 7. title of plot in left panel is unnecessary
121. acronym "pe" never defined (see line 116)
Equation 3. \mathrm{DCR}, subscripts uptight, add coma at the end.
Figure 9. Units should be uptight (e.g., \mathrm{kHz / V})
156. remove word "reunión"
Equation 4. add coma at the end
157. rephrase: "where N^\mathrm{A}_\mathrm{1~pe} and N^\mathrm{B}_\mathrm{1~pe} are the number of events above 0.5 pe after and before stimulation, respectively."
157. usage of T for time is confusing. T^A never used afterwards...
162. "Figure 10 shows... -> "In the right panel of Figure 10 ... are shown"
Equation 5. add coma at the end.
Figure 10. MuTe-SiPM -> SiPM. Also, refer to left and right panels in the same way as in previous figures.
167. the afterpulsing probability -> the probability of afterpulsing
167. the crosstalk below -> of crosstalk is below
4. OPERATING TEMPERATURE CONDITIONS OF MUTE
175. of the MuTe -> of MuTe
179. Macin -> Machin
179. Cerro-Machin -> Cerro Machin
179. at the Cerro-Macin the average temperature is -> the average temperature at Cerro-Machin is
181. remove comas
182. comes from -> occurs between
191. Figure 11 displays -> In Figure 11 ... is shown
5. TEMPERATURE INNFLUENCE ON THE MUTE-SIPMS
199. above -> in Section 4.2
202. stars -> starts
202. at the 00:00 -> at 00:00h
203. at morning (06:30) -> at 06:30h
204. at day (13:00) -> at 13:00h
Figure 13. Left plot: V_{b} -> V_\mathrm{br}
Figure 13. Missing reference to Left and Right plots in caption
Figure 13. of typical temperature values -> of the temperature
212. rounds -> is approximately
216. = -> is
216. "reuniónout"
217. (Right) -> (right)
218. (Left) -> (left)
Figure 14. Missing reference to Left and Right plots in caption
Figure 14. for typical temperature values -> with temperature
220. In Figure 15 we show... -> In Figure 15 ... is shown.
221. The MuTe was set... -> For this measurement, MuTe was set...
Figure 15. Is there any explanation or comment about the spike in rate between day 24 and 25?
230. of -0.057 Hz / ºC
234. testes -> tests
239. On the other hand -> at the same time, we found that the DCR...
243. the probabilities of afterpulsing and crostalk...
252. amd -> and

Author Response

(The authors gave the same response as above.)
